# Relationship between Dysphagia and Home Discharge among Older Patients Receiving Hospital Rehabilitation in Rural Japan: A Retrospective Cohort Study

**DOI:** 10.3390/ijerph191610125

**Published:** 2022-08-16

**Authors:** Ryuichi Ohta, Emily Weiss, Magda Mekky, Chiaki Sano

**Affiliations:** 1Community Care, Unnan City Hospital, 96-1 Iida, Daito-cho, Unnan 699-1221, Japan; 2Department of Public Health, Old College, University of Edinburgh, South Bridge, Edinburgh EH8 9YL, UK; 3Department of Community Medicine Management, Faculty of Medicine, Shimane University, 89-1 Enya-cho, Izumo 693-8501, Japan

**Keywords:** dysphagia, home discharge, older adults, rural hospital, rehabilitation, temporal flat lateral position, polypharmacy

## Abstract

Dysphagia refers to swallowing difficulty, which impacts patients’ quality of life. Dysphagia influences clinical outcomes, including mortality rates and length of hospital stay of older hospitalized patients. Dysphagia may affect the current and future quality of life of these patients. However, its exact impact remains unclear. We aimed to clarify the impact of dysphagia on discharge to home in older patients in a rural rehabilitation unit. We conducted a secondary analysis using data from a retrospective cohort study including patients aged over 65 years who had been discharged from a community hospital rehabilitation unit in rural Japan. Data from the participants had been previously collected from April 2016 to March 2020. The primary outcome was home discharge. The average participant age was 82.1 (standard deviation, 10.8) years; 34.5% were men. Among medical conditions, brain stroke (44.3%) was the most frequent reason for admission; the most frequent orthopedic condition was femoral fracture (42.9%). The presence of dysphagia (odds ratio [OR] = 0.38, 95% confidence interval [CI]: 0.20–0.73), polypharmacy (OR = 0.5, 95% CI: 0.32–0.90), and admission for internal medicine diseases (OR = 0.44, 95% CI: 0.26–0.77) were negatively associated with home discharge. High motor domain scores of the Functional Independence Measure were positively associated with home discharge (OR = 1.07, 95% CI: 1.05–1.08). Dysphagia was negatively associated with home discharge as were polypharmacy and admission for internal medicine diseases and conditions. By clarifying effective interventions through interventional studies, including approaches to managing multimorbidity and polypharmacy through interprofessional collaboration, the health conditions of older patients in rural areas may be improved.

## 1. Introduction

### 1.1. Dysphagia in Older Patients

Dysphagia is defined as the condition in which people cannot swallow liquid or solid food because of mechanical or functional problems; it is a critical issue in older patients and impinges on their lives [1]. Among older people, various severe diseases, such as cardiovascular diseases and frailty, cause dysphagia [2,3]. A common condition causing dysphagia is brain infarction, where blood loss to parts of the brain results in cell death, leading to neurological abnormalities in the movement of pharyngeal and laryngeal structures [4,5,6,7,8]. Heart failure and other chronic diseases such as diabetes and acute infections can also cause dysphagia because of the increasing frailty associated with the process of disease progression [7,9,10].

Furthermore, older patients tend to have multiple chronic diseases, which can increase the risk of developing dysphagia [11]. Based on previous studies, the prevalence of dysphagia is 30–40% among hospitalized patients aged > 65 years worldwide [3,10]. Aging can cause a deterioration in the physical and psychological conditions, leading to the progression of disuse syndrome [1]. This deterioration impinges on the quality of life of older people and has been associated with difficulties in the continuity of home care and an increase in mortality rates [3,12,13].

In Japan, a country with one of the largest older populations worldwide, diseases associated with high mortality, such as pneumonia and aspiration pneumonia, following cancer, heart diseases, and cerebrovascular diseases, have become more prevalent [14]. Previous studies have shown that 70% of patients with pneumonia are more than 75 years old and are diagnosed with aspiration pneumonia [15].

Furthermore, more than 20% of patients older than 75 years with dysphagia die within 2 months after hospital admission [15]. Moreover, 50% of patients with both pneumonia and dysphagia die within one year after hospital admission [16]. Thus, dysphagia is significantly related to mortality among older patients, thereby making it necessary to sustain and improve their ability to swallow using appropriate rehabilitation. Because the progression of dysphagia can be triggered by various factors, implementing interventions to address dysphagia in aging societies can be difficult.

### 1.2. Swallowing Mechanism

Swallowing is performed by various muscles in the neck innervated by peripheral and central nerves. The major muscles involved in swallowing are the suprahyoid muscles (digastric, mylohyoid, geniohyoid, and stylohyoid muscles) [17]. These muscles act in coordination to lift the larynx and obstruct the pharynx, enabling humans to swallow. Swallowing can be classified into the following five phases: recognition, oral preparatory, oral transport, pharyngeal transport, and esophageal transport [17].

Owing to cognitive function decline, swallowing training can be challenging [2,18,19]. Appropriate treatment and rehabilitation can be performed by judging which swallowing phases are impaired [20]. Aging can weaken the pharyngeal muscles, which can severely influence swallowing ability. Thus, reclining positions can increase the risk of food entering the larynx and, consequently, the risk of aspiration [18,21].

The progression of dysphagia can impact the continuity of home care, leading to an increase in the duration of hospital admission and exhaustion of hospital medical professionals [4,9,22]. Thus, sustaining the ability to swallow can improve the possibility of safe long-term home care for older people with dysphagia [20]. Moreover, swallowing methods that do not employ a reclined position may enable such patients to sustain swallowing. Therefore, new swallowing methods should be developed to improve the quality of life of older patients with severe dysphagia.

### 1.3. Relationship between Dysphagia and Health Outcomes

The presence of dysphagia affects outcomes of older hospitalized patients, such as mortality and length of hospital stay [6,10]. Dysphagia is induced by physical frailty, which can be exacerbated by hospitalization [23] since hospitalization results in the deterioration of muscle and physical function in patients confined to bed [3,23]. The impact of hospitalization is more substantial among older people, and these patients can lose their ability to swallow [3,23,24]. Studies conducted at hospitals specializing in rehabilitation have shown that dysphagia affects the duration of hospital stay because caregivers experience difficulties in feeding patients [12,25,26].

Furthermore, dysphagia can increase readmission to hospitals because of the high possibility of aspiration and immunological impairment [12,25,26]. These problems have become common worldwide [1]. The evidence regarding dysphagia has been gathered mainly from older hospitalized patients with heart failure, brain stroke, dementia, and aspiration pneumonia [10,11,12,25]. Older hospitalized patients with these diseases who have dysphagia tend to have a longer hospital stay and are discharged to long-term care facilities instead of their homes [10,11,12,25].

### 1.4. Difference in Rehabilitation between Urban and Rural Settings

Dysphagia may affect the present and future quality of life of hospitalized patients. The interventions for dysphagia are administered by speech therapists based on various swallowing tests [27]. Based on the evidence, early assessment and intervention can improve the outcomes of older patients with dysphagia [3,15,20]. As older hospitalized patients tend to become frail, interventions for dysphagia should begin as early as possible. The rehabilitation outcomes of older people during hospitalization and after discharge depend on the hospital and community conditions such as rural or urban settings [28,29].

Rehabilitation can be performed in various situations, and the typical differences in the quality of rehabilitation are largely predicted by whether the hospital location is urban or rural [28,29]. In rural settings, few hospitals specialize in rehabilitation [28,29,30]. Rehabilitation is commonly performed in community hospitals that accommodate patients with a wide variety of conditions and diseases from emergency to chronic conditions [22].

Owing to a rapidly aging population and decreasing birth rate, rural and urban hospitals have to deal with frailty and dysphagia problems, which can be caused by aging [23,31]. Since few studies have been conducted in rural hospitals, there is an evidence gap between urban and rural hospitals regarding rehabilitation [1,3,28,29,32].

### 1.5. Rationale and Purpose of This Study

Dysphagia affecting the outcomes of older hospitalized patients in rural hospitals is a critical problem because of the drastically aging population. The optimal outcome for hospitalized patients is home discharge [1,3]. Home discharge supported by effective interprofessional care gives hope to patients and eases the burden on health insurance, which should be encouraged for sustainable medical care [18,33].

At the time of writing, no study had assessed older hospitalized patients with dysphagia in rural hospitals. By clarifying the current condition of dysphagia and its impact on the rate of home discharge, the obtained evidence can inform the intervention for dysphagia in older hospitalized patients in rural areas. This study aimed to clarify the impact of dysphagia on home discharge in older hospitalized patients in a rural rehabilitation unit.

## 2. Materials and Methods

### 2.1. Purpose

This study’s purpose was to clarify the relationship between the presence of dysphagia among older patients and discharge to their home from a rural rehabilitation unit.

### 2.2. Design

This was a retrospective cohort study using data from patients aged over 65 years who had been discharged from a rehabilitation unit of a community hospital in rural Japan (Unnan City Hospital). The data from the participants had been previously collected from Unnan City Hospital’s digital medical record.

### 2.3. Setting

Unnan City is one of the most rural cities in Japan, located in the southeast section of the Shimane Prefecture. In 2020, the total population of Unnan was 37,638 (18,145 men and 19,492 women). Thirty-nine percent of the population are aged over 65 years; and this number is expected to reach 50% by 2050 [34]. There are 16 clinics, 12 home care stations, 3 visiting nurse stations, and 1 public hospital (Unnan City Hospital).

The hospital staff comprises 27 physicians, 197 nurses, 7 pharmacists, 15 clinical technicians, 37 therapists (22 physical therapists, 12 occupational therapists, and 3 speech therapists), 4 nutritionists, and 34 clerks. There is no other medical institution with a recovery rehabilitation unit within the city.

### 2.4. Recovery Rehabilitation Unit

The recovery rehabilitation unit of the Unnan City Hospital had 30 rehabilitation beds during the study period. The unit accommodated patients who were motivated to return home after their rehabilitation. The underlying diseases requiring rehabilitation were mainly in the domains of internal medicine and orthopedics.

Individual patients and their home settings—such as living alone or with their families—were discussed by the physician and chief nurse (who were responsible for the recovery rehabilitation unit) together with the patients and their families. The decision to move from acute care to the recovery rehabilitation unit was made collaboratively. Rehabilitation therapy was performed on average two times per day (60–90 min per session) by the physical and occupational therapists. Speech therapists were included if the patient had swallowing and speaking problems.

The discharge timing and place were selected based on discussions involving patients, their families, a specific team consisting of the physician and nurse in charge, and social workers. The team facilitated the discussion regarding support and decision-making for patient discharge. Since Japanese health insurance covers hospital rehabilitation for 150 days, most patients admitted to rehabilitation units use most of the duration that is covered by their insurance.

### 2.5. Participants

Data from patients aged over 65 years who were discharged from the Unnan City Hospital after the treatment of acute diseases and training in the rehabilitation unit were used for secondary analysis in this study. Data from all consecutive patients who were discharged during the four years from 1 April 2016 to 31 March 2020, and those who provided informed consent were included in this study. When consent could not be obtained from the patients who lacked the capacity to fully understand the purpose of the study, consent was obtained from the next of kin.

Each year, the rehabilitation unit accommodates approximately 200 patients. Because of the high prevalence of dysphagia among older people, all patients are checked upon admission regarding their capacity to swallow. In this study, the patients were divided into two groups: those with and without dysphagia.

#### 2.5.1. Inclusion Criteria

-Aged over 65 years and discharged from the Unnan City Hospital after treatment of acute diseases and training in the rehabilitation unit-Discharged between 1 April 2016 and 31 March 2020-Consented to their data being used in this research

#### 2.5.2. Exclusion Criteria

-Lacked key outcomes and predictors: body mass index (BMI), blood albumin concentration, care level, duration of rehabilitation, dysphagia, and the discharge conditions.-Admitted to the rehabilitation unit and died or transferred to the acute unit in the hospital during the admission because of acute changes in their conditions.

### 2.6. Measurements

Patient information was extracted from the electronic medical records of the Unnan City Hospital. The hospital medical clerks extracted the data, anonymized them after extraction, and gave the primary researcher the anonymized dataset. The anonymized datasets are owned by the Unnan City Hospital. The extracted, anonymized data were stored on the researcher’s Unnan City Hospital secure network drive as password-protected files. They were not linked in any way to the participant consent forms or full dataset.

#### 2.6.1. Primary Outcome

The main outcome was patient discharge to home. The discharge timing and place (home or long-term care facilities) were selected by the multidisciplinary team working closely with the patient and the family considering the quality of life of the patient and the burden on the family [22].

#### 2.6.2. Independent Variable

The presence of dysphagia was defined based on the diagnosis by the otorhinolaryngologist using a water swallowing test (WST) score of three points or less. All patients were assessed by nurses for their abilities to swallow using the WST for the screening of dysphagia. The method used to diagnose dysphagia and perform swallowing rehabilitation was established based on the results of the videoendoscopic test (VE) and videofluoroscopic examination (VF). If the WST score was three or less, the patients were assessed by VE performed by an otorhinolaryngologist to establish the dysphagia diagnosis and the rehabilitation procedure. If there were delays in swallowing or dysfunction of swallowing with no confirmed aspiration, speech therapists rehabilitated patients. If there was confirmed aspiration, VF was performed to determine what food consistency the patients could eat safely. Nurses and care workers helped patients with dysphagia to eat their food. Two groups of patients were distinguished by the presence or absence of dysphagia based on VE or VF.

#### 2.6.3. Covariates

The following variables were extracted from the medical records at admission: age (years); sex (male or female); BMI (kg/m^2^); serum albumin level (g/dL) as an indicator of nutrition status; reasons for admission (confirmed disease names or symptom names); number of medications to assess polypharmacy at admission [35]; the Charlson Comorbidity Index (CCI) calculated from the patient medical histories which shows the severity of the patient’s medical conditions [36]; care level based on the Japanese long-term insurance system (numbered from 1 to 5, with 1 being the least dependent and 5 being severely dependent); cognitive and motor components of the Functional Independence Measure (FIM) at admission which are measured by therapists as an indicator of patient activity in daily life; and discharge destinations (home or facility). The reasons for admission were categorized as orthopedic and medical conditions (Table 1).

##### Polypharmacy

Polypharmacy is defined as the use of multiple medicines by one patient for various diseases. The number of drugs that define polypharmacy varies across studies. Depending upon the study, the use of more than five or six drugs can be categorized as polypharmacy and can be associated with various complications, such as aspiration pneumonia, femoral neck fracture, dizziness, and dementia [35,37,38,39]. Such complications can cause difficulties in terms of discharge to home because of the risk of readmission. In this study, we used the definition of polypharmacy as the number of drugs exceeding five because the use of more than five medications can increase mortality and long-term facility discharge [35].

##### Charlson Comorbidity Index

The CCI was established for the classification of prognostic comorbidity based on the age and past medical history of the patient. This score has been used for adjusting the confounding influence of comorbid conditions on overall survival. Based on previous studies, the CCI score is related to death from comorbid diseases [36].

In this index, the following factors are used to create the score: a weight of one for myocardial infarction, congestive heart failure, peripheral vascular disease, cerebrovascular disease, dementia, chronic pulmonary disease, connective tissue disease, peptic ulcer disease, mild liver disease, and diabetes; a weight of two for hemiplegia, moderate or severe renal disease, diabetes with end-organ damage, and any malignancy; a weight of three for moderate and severe liver disease (e.g., cirrhosis with ascites); and a weight of six for metastatic solid tumor or acquired immunodeficiency syndrome. Age is scored as follows: 50 to 59 years, 1 point; 60 to 69 years, 2 points; 70 to 79 years, 3 points; and more than 80 years, 4 points. The sum of the age and comorbidity scores define the total CCI score [36].

##### Care Level Based on the Japanese Long-Term Care Insurance System

Care level is selected based on the Japanese long-term insurance system. There are six care levels: level 0 (needing care for less than 32 min/day), level 1 (needing care for 32–50 min/day), level 2 (needing care for 50–70 min/day), level 3 (needing care for 70–90 min/day), level 4 (needing care for 90–110 min/day), and level 5 (needing care for more than 110 min/day).

In the Japanese long-term care insurance system, the estimated time of needed care is calculated through the formula produced by the government using questionnaires for the level of dementia (including the symptoms of Behavioral and Psychological Symptoms of Dementia) and activities of daily living (ADL; how much help is needed directly or indirectly, in terms of regular life tasks or medical care), completed by primary care physicians and governmental clerks from local governments. Based on the estimated time, each local government holds a meeting among various healthcare professionals in cities or towns including physicians, dentists, public health nurses, and care managers. Care levels for individuals are decided through discussion in these meetings [40].

##### Functional Independence Measure

The FIM is used for the assessment of basic ADL in patients in various situations. The FIM score is related to home discharge. The FIM consists of 18 items and is categorized into two components: motor and cognition. The scores for the two components are calculated based on the seven grades in each subscale. The motor subscales consist of eating, grooming, bathing, dressing the upper body, dressing the lower body, toileting, bladder management, bowel management, transfers to bed/chair/wheelchair, transfers to the toilet, transfers to the bath/shower, walking/using a wheelchair, and using the stairs. The cognition subscales consist of comprehension, expression, social interaction, problem solving, and memory. Each item is scored on seven original scales using a score of 1 to 7 (Table 2). A higher score indicates a more independent status. The range of motor FIM score is 13 to 91. The cognitive FIM score ranges from 5 to 35 [41].

### 2.7. Statistical Analysis

The reasons for admission are presented demographically with the categories of specific disease names. The data are descriptively summarized by dividing the participants into two groups: those with and without dysphagia. Based on the test of normality, Student’s *t*-test was performed on parametric data, and the Mann–Whitney U test on nonparametric data.

Based on previous studies and the average of the variables, numerical variables were parsed as follows: CCI score (≥5 and <5) [36], care level (≥1 and <1) based on the burden on caregivers and families [40], and the number of medicines (≥5 and <5) as polypharmacy [38].

The primary outcome of discharge to home was a binary variable. Therefore, to investigate the relationship between dysphagia and discharge to home, a logistic regression model was used with variables reported to be significantly associated with dysphagia and home discharge in previous studies [30,41,42]. The previous studies showed that discharge to home is associated with age, sex, CCI score, care level, polypharmacy, albumin concentration, the reason for admission, and FIM score [30,41,42]. Referring to previous research, this study used a logistic regression model with forced entry and included all the previous independent variables. For the model construction, C-statistic was calculated to check the model performance.

Regarding the sample size calculation, 752 participants were determined to provide 80% statistical power and 5% type 1 error to detect a difference in the percentage of patients with home discharge of 10% between the dysphagia and no dysphagia groups. Statistical significance was defined as *p* < 0.05. All statistical analyses were performed using EZR version 1.51 (Saitama Medical Center, Jichi Medical University, Saitama, Japan; URL: http://www.jichi.ac.jp/saitama-sct/SaitamaHP.files/OSXEN.html, accessed on 24 March 2022), a graphical user interface for R (R Foundation for Statistical Computing, Vienna, Austria) [43].

### 2.8. Assessment of Ethical or Other Risks and Permissions

Ethical approval was obtained from The University of Edinburgh Ethics Committee (Approval ID: AC21138, Approval date: 18 January 2022) and the Unnan City Hospital Ethics Committee (Approval ID: 20200030, approval date: 1 November 2021). Data were stored and analyzed in accordance with The University of Edinburgh guidelines.

## 3. Results

### 3.1. Participant Selection

Figure 1 shows the flowchart of the study population selection process. In total, 951 patients were admitted to the rehabilitation unit between 1 April 2016 and 31 March 2020. Among them, 845 were over 65 years. Sixty-two patients were excluded because of death during admission and the lack of data regarding the covariates as follows: excluded due to death during admission, *n* = 9 and excluded due to missing data, *n* = 53 (BMI, *n* = 16, albumin, *n* = 14, care level, *n* = 13, and duration of rehabilitation, *n* = 10) (Figure 1).

### 3.2. Demographics of the Participants

The average age of the participants was 82.12 (standard deviation = 10.77) years, and the proportion of men was 34.5%. Between the groups with and without dysphagia, there were significant differences in the following factors: sex (*p*-value < 0.001), reasons for admission (*p*-value < 0.001), home discharge (*p*-value < 0.001), FIM score at admission (total [*p*-value < 0.001], motor [*p*-value < 0.001], and cognitive scores [*p*-value < 0.001]), care level (*p*-value = 0.024), CCI score (*p*-value < 0.001), CCI ≥ 5 (*p*-value = 0.003), and the presence of liver diseases (*p*-value = 0.016) (Table 3). Between the groups with and without dysphagia, there were no significant differences in the following factors: age (*p*-value = 0.684), albumin (*p*-value = 0.305), BMI (*p*-value = 0.181), and dependent condition (*p*-value = 0.234) (Table 3).

### 3.3. Reasons for Hospital Admission

Table 4 lists the diagnoses of patient conditions and their frequencies for hospital admission in the medical and orthopedics categories. In the medical category, brain stroke (44.3%), followed by brain hemorrhage (23.9%), and pneumonia (6.7%) were the most frequent conditions. Among the orthopedic conditions, femoral fracture (42.9%), followed by compression fracture (26.0%), and knee osteoarthritis (10.7%) were the most frequent conditions.

### 3.4. Relationship between Dysphagia and Home Discharge

The multivariate regression model was used to investigate the relationship between dysphagia and home discharge. Referring to previous research, this study used the logistic regression model with forced entry by including all the previous independent variables. The C-statistic of the regression model was 0.901 (95% confidence interval [CI]: 0.873–0.929). In the results of the multivariate logistic regression model, the presence of dysphagia (odds ratio [OR] = 0.33, 95% CI: 0.20–0.73), polypharmacy (OR = 0.5, 95% CI: 0.32–0.90), and admission for diseases of internal medicine (OR = 0.44, 95% CI: 0.26–0.77) were negatively associated with home discharge. A high motor domain score of the FIM was positively associated with home discharge (OR = 1.07, 95% CI: 1.05–1.08). Meanwhile, other factors such as age, sex, serum albumin, cognitive domain score of FIM, dependent conditions, and CCI were not associated with home discharge (Table 5).

## 4. Discussion

For effective discharge to home, approaches to dysphagia should be considered in older patients, considering polypharmacy and ADL, as this study shows. Due to the negative effect of polypharmacy, the number and composition of medicines used by the patient should be noted upon admission, with the aim of reducing the number of medicines prescribed, especially in the internal medicine category. Moreover, based on the present and previous research, improving and sustaining ADL during admission may be essential for effective rehabilitation that leads to home discharge [30,41,42].

### 4.1. Dysphagia Impinges on Home Discharge of Older Patients in Rural Areas

Dysphagia leads to the deterioration of the patient’s medical condition. This research shows that dysphagia was related to difficulties in discharging patients to home. Within the hospital, patients with dysphagia needed special care around food consumption, with support from medical staff such as nurses, therapists, and care workers. Patients who cannot consume enough food by swallowing may need feeding using a transient tube to ensure proper nutrition [44]. Their rehabilitation quality can decrease in the rehabilitation period, and they become frailer than patients who do not have dysphagia [42,43]. This frailty can cause deterioration in health conditions and slow down their improvement [42,43].

Dysphagia also increases the burden on home medical care staff and caregivers, which directly impacts their ability to return home. There are various types of evidence supporting our results. The support needed for dysphagia can result in a substantial burden on home medical care staff [44]; for example, preparing and giving food to patients with dysphagia requires specific knowledge and skills [45]. Although home medical care staff are trained to provide this type of support, the burden associated with caring for individuals with dysphagia is still substantial and can directly affect individuals’ home discharge [46]. The context of our research could affect the results. In rural areas, the number of home care professionals is limited. Thus, few home care workers are available to sufficiently support patients with dysphagia in their homes.

Furthermore, home caregivers are not accustomed to providing such support [46]. Previous studies have shown that the burden on caregivers providing support for dysphagia can be significant in a home care setting [47]. Caregivers have to learn methods of providing support for dysphagia [47,48,49]. In this study, the discharge place was decided through discussion between medical professionals, patients, and home care givers. Workload and mental stress on the home caregiver can influence the home discharge rate among older patients in this study’s results. Older individuals living by themselves have also to perform various activities independently [49]. If older individuals in rural areas have diseases that require hospital admission, they have to recover sufficiently because there may not be family members nearby to provide support at home [50,51].

### 4.2. Medicines Affecting Dysphagia in Internal Medicine

Various medicines can cause dysphagia, especially drugs such as those used to treat depression and insomnia. This research shows the possibility that several medicines can be associated with dysphagia. A previous study carried out in rural contexts showed the negative effects of medicines for mental illness on discharge to home [22]. The association can be explained by symptoms and diseases in the elderly. Many older patients have various symptoms because of multiple diseases [52,53,54]. Some of them need medicines for symptoms of their chronic symptoms [53,54]. These medicines can affect the patients’ swallowing abilities because of dry mouth and drowsiness [55]. As saliva and being alert are essential for swallowing, patients using these medicines may have difficulty swallowing [56,57,58]. Medicines for multimorbidity can cause and exacerbate dysphagia in older admitted patients.

Older patients admitted under internal medicine tend to have multimorbidity and require comprehensive care from various professionals and multiple medicines to control their symptoms. This approach to multimorbidity can cause polypharmacy, relating to dysphagia [59,60,61]. This study also shows that admission for the reason of internal medicine could impinge on discharge to home. To moderate the risk of polypharmacy and delirium, the involvement of various medical professionals is vital.

### 4.3. ADL Affecting Discharge to Home

This study clarified the contribution of motor FIM to discharge to home care, and cognitive FIM was related to their discharge to home. To decide upon the suitability of discharge to home, not only the change of motor FIM after rehabilitation but also basic physical ADL after the onset of diseases could be related [22]. At home, older people must sustain life functions regarding ADL, which are enhanced through rehabilitation. Cognitive functions may also affect older patient lives in their home to manage their usual lives. This research shows that the average cognitive FIM component was 32, higher than the suggested standard of living at home [34].

The study shows no relationship between discharging to homes and cognitive FIM statistically. Moreover, this study’s participants’ motor FIM with dysphagia was 61 on average, showing that they are frail [19,25]. More than 50% of them could go back to their home from the hospital. This research reveals a high possibility of older frail patients returning to their homes. Among older patients with cognitive dysfunction, their motor FIM improvement enables them to be discharged to their home. This study’s participants are collected only from rehabilitation wards. Future studies can clarify the cause-and-effect relationship between improvement of the FIM and dysphagia, and discharge to home among various older patients in rural hospitals.

### 4.4. Factors Not Related to Home Discharge among Older Patients

Age might not be related to home discharge among older admitted patients. This study shows that aging is not associated with home discharge. This study focused on older generations, and their differences in physical and cognitive abilities may not differ. In addition, this study’s population is approximately 80 years old, so the change in their functions could vary, and the factor of age may not show a direct significant effect on discharge to their home from the hospital.

The difference in sex could not be related to home discharge. As this study shows, sex difference did not contribute significantly to home discharge. In lay care usage such as self-management, consulting with families and friends, and over-the-counter drugs, there is no difference in the frequency of care between men and women [62,63]. Meanwhile, women tend to be associated with frequent usage of emergency departments [64]. These trends can be associated with the stereotype in some countries that men should be strong and masculine [63,64]. Moreover, in the primary care setting, there are studies with different results involving usage rates between men and women who have critical symptoms [65,66,67,68]. In rural contexts, primary care physicians could find older people’s critical symptoms and refer them to general hospitals for further investigation and hospitalization [67,68]. Therefore, sex differences may not negatively affect their clinical courses of critical diseases in communities and hospitals in this study.

This study shows that serum albumin may not be related to home discharge. Serum albumin levels may be associated with emergency admission to hospitals among older patients. Serum albumin levels may also be associated with infection, dehydration, and nutritional deficiency [4,69]. In contrast, regarding older generations over 80 years old, there is a lack of evidence regarding the relationship between nutritional levels and home discharge. Their home discharge could be strongly affected by social conditions and their physical abilities. Therefore, this study may not show the relationship between home discharge and serum albumin levels.

In this study, the dependent conditions of the participants were not related to home discharge among the participants. Patients with dependent conditions have various kinds of diseases and may be exposed to high risks of acute conditions in their homes, which may impinge on their lives at home [70]. Medical staff consider that patients with dependent conditions should be cared for intensively. As their activities may not be intensive, they may not have many opportunities to develop symptoms, such as chest pain, dyspnea, and joint pain [71]. Additionally, as they may not be exposed to a range of people except medical staff, they may not frequently have communicable diseases [70]. Instead, less dependent patients may show symptoms according to their activities. As they may have certain problems expressing their symptoms, the various expression of symptoms may make their care givers anxious at home [48]. In contrast, the ability to express symptoms could facilitate early detection of critical symptoms, which can facilitate effective home care of less dependent patients.

This study shows that CCI may not have a statistical relationship with home discharge. Past medical history can be associated with an individual’s medical usage, triggered by symptoms from their diseases. People with specific chronic diseases and high CCI tend to use professional care because they may be accustomed to the usage of medical care for their frequent acute symptoms [72]. They can approach medical care quickly, allowing their medical conditions to be treated before becoming critical. In contrast, patients with low CCI may be able to control their symptoms with self-management [67] and may try to manage their symptoms with usual self-care. When they have symptoms with alarming signs, they may access medical care quickly, possibly leading to effective medical care. Therefore, high and low CCI could contribute to quick and effective medical care, and this study may not show the relationship between CCI and home discharge.

### 4.5. Comprehensive Methods to Address Dysphagia among Older Patients

The approaches to the issues of dysphagia in older patients may need multiple interventions, including controlling multimorbidity and reducing medicines. Controlling multimorbidity in admitted older patients with acute diseases requires patient assessment by various professionals. In this study, the support for patients with dysphagia was performed only by therapists, nurses, and care workers. In patient care, physicians must diagnose patient diseases effectively and treat them with appropriate medicine and interventions involving patient and family participation [73]. Nurses, pharmacists, and social workers can assess patient conditions in the context of patient discharge, considering the need for home care and reducing medicines [74,75,76,77]. As found in this study, older patients who were admitted may suffer from various diseases and need adjustment of medicine and care in their home. The collaboration among all these professionals can improve patient care, including dysphagia, and enable them to live in their homes.

Reducing the rate of polypharmacy to prevent the deterioration of swallowing in older patients needs collaboration among multiple professionals. Based on the assessment of polypharmacy by pharmacists, physicians can choose which medicines can be reduced [78]. As this study shows, polypharmacy may affect the discharge destination from hospitals for elderly patients. Approaches to polypharmacy can mitigate the negative effect of multiple medicines, leading to better conditions for the rehabilitation of swallowing by reducing the possibility of medicine-related complications [78].

Various swallowing methods can be applied for the continuation of oral food intake. The consistency of food can be changed according to the ability to swallow. Based on this study, there are many patients with brain stroke and hemorrhage who might have poor swallowing abilities because of neurological abnormalities. Orthopedic patients may also become frail owing to the limitation of their movement because of fractures [2,3], and their frailty can cause dysphagia [2,3]. Changing the consistency of food to a semi-solid or liquid form can enable patients to consume their food smoothly, lowering the risk of aspiration [79]. In addition, new eating methods for patients with dysphagia has been invented in developed countries [79]. By using the method, older patients with severe dysphagia and those who cannot consume food in conventional ways may be able to eat normally. The method could be also effective even in rural hospitals and rehabilitation units [79]. Future studies should investigate comprehensive approaches in these patients, including education on new swallowing methods for medical professionals, patients, and their families.

### 4.6. Limitations

This study was performed at a single rehabilitation center in a rural Japanese hospital, and this might have affected the external validity. Future studies should investigate the effect of dysphagia on home discharge in older patients admitted to different types of hospitals in other countries. Nevertheless, this study can be used as a foundation for the investigation of rural rehabilitation regarding dysphagia.

Second, the excluded patients could have influenced the results. Excluded patients who died during admission may have had more severe conditions. BMI indicates a patient’s nutritional condition and is usually measured upon admission. Patients who lacked these data could have been more dependent because they were possibly unable to stand upright for their weight and height to be measured to calculate their BMI. Moreover, patients without albumin concentration data could have been in good nutritional condition because this measurement may not have been taken owing to their nutritional status. Therefore, the elimination of these participants may have affected the nutritional statistics in this study. Furthermore, care level and the duration of rehabilitation may have affected rehabilitation intensity. Patients without these data may have been more independent and, hence, may not have required assessment in terms of care level and may have been discharged to home without challenges. Thus, the reliability of the results of this study may have been affected. However, the number of excluded participants was relatively low; hence, the reliability ought to be retained.

## 5. Conclusions

The presence of dysphagia can negatively affect the discharge to home among patients admitted to a rehabilitation unit in a rural Japanese hospital. This relationship is a critical issue for aging societies in terms of sustaining comprehensive care for older people. The presence of dysphagia may be negatively associated with home discharge as are polypharmacy and admission for diseases treated under the umbrella of internal medicine. For effective discharge to home, approaches to alleviate dysphagia should be considered for rural older patients.

## Figures and Tables

**Figure 1 ijerph-19-10125-f001:**
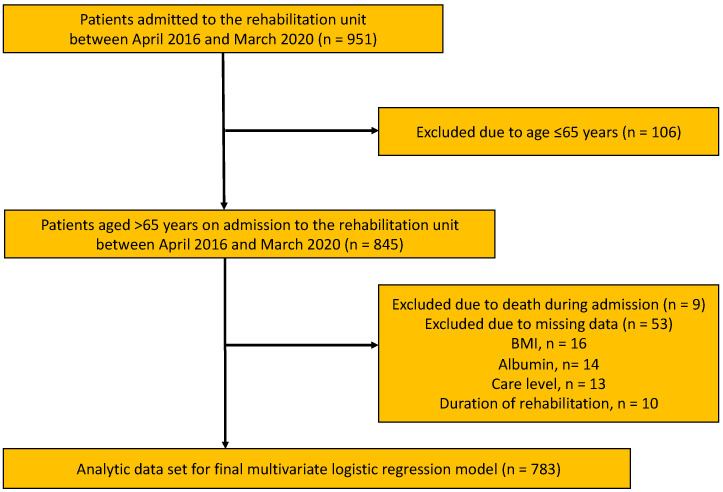
Flowchart of the participant selection. BMI, body mass index.

**Table 1 ijerph-19-10125-t001:** Definitions of the variables.

Variables	Definitions
Primary outcome	Discharge destination (Home or long-term care facilities)
Patient demographics	Age (years)
	Sex (male or female)
	Body mass index (kg/m^2^) at admission
	Serum albumin (g/dL) at admission
	Number of medications at admission
	Charlson Comorbidity Index score at admission
	Care level based on the Japanese long-term insurance at admission
	Cognitive and motor component scores of the Functional Independence Measure at admission Reasons for admission (orthopedic or medical condition)

**Table 2 ijerph-19-10125-t002:** Standards of each score for the Functional Independence Measure items.

Standards	Explanation
7	Complete independence (timely, safety)
6	Modified independence (with device)
5	Supervision (subject = 100%)
4	Minimal assistance (subject = 75% or more)
3	Moderate assistance (subject = 50% or more)
2	Maximal assistance (subject = 25% or more)
1	Total assistance or not testable (subject less than 25%)

**Table 3 ijerph-19-10125-t003:** Demographics of the participants.

		Dysphagia		
Factor	Total	Yes	No	*p*-Value
N	783	101	682	
Age (years), mean (SD)	82.12 (10.77)	81.71 (8.87)	82.18 (11.03)	0.684
Male sex (%)	270 (34.5)	53 (52.5)	217 (31.8)	<0.001
Albumin (g/dL), mean (SD)	3.76 (0.56)	3.70 (0.58)	3.77 (0.56)	0.305
BMI (kg/m^2^), mean (SD)	21.37 (4.91)	20.75 (3.81)	21.47 (5.04)	0.181
Medicines taken, mean (SD)	4.91 (3.00)	5.45 (3.44)	4.83 (2.93)	0.055
Patients with polypharmacy, *n* (%)	402 (51.3)	57 (56.4)	345 (50.6)	0.288
Reasons for admission, *n* (%)				
Medical	314 (40.1)	72 (71.3)	242 (35.5)	<0.001
Orthopedic	469 (59.9)	29 (28.7)	440 (64.5)	
FIM score at admission				
Total FIM score (median)	109.00 (13.00, 126.00)	86.00 (18.00, 126.00)	111.00 (13.00, 126.00)	<0.001
Motor domain score (median)	78.00 (2.00, 91.00)	61.00 (13.00, 91.00)	79.00 (2.00, 91.00)	<0.001
Cognitive domain score (median)	32.00 (0.00, 35.00)	28.00 (5.00, 35.00)	32.00 (0.00, 35.00)	<0.001
Discharge to home (%)	643 (82.1)	54 (53.5)	589 (86.4)	<0.001
Care level (%)				
0	567 (72.4)	68 (67.3)	499 (73.2)	0.024
1	39 (5.0)	1 (1.0)	38 (5.6)	
2	72 (9.2)	10 (9.9)	62 (9.1)	
3	48 (6.1)	8 (7.9)	40 (5.9)	
4	29 (3.7)	6 (5.9)	23 (3.4)	
5	28 (3.6)	8 (7.9)	20 (2.9)	
Dependent condition (%)	216 (27.6)	33 (32.7)	183 (26.8)	0.234
CCI score (%)				
2	40 (5.1)	1 (1.0)	47 (6.9)	<0.001
3	66 (8.4)	7 (6.9)	59 (8.7)	
4	209 (26.7)	20 (19.8)	189 (27.7)	
5	163 (20.8)	24 (23.8)	139 (20.4)	
6	150 (19.2)	24 (23.8)	126 (18.5)	
7	80 (10.2)	14 (13.9)	66 (9.7)	
8	37 (4.7)	4 (4.0)	33 (4.8)	
9	20 (2.6)	5 (5.0)	15 (2.2)	
10	9 (1.1)	2 (2.0)	7 (1.0)	
12	1 (0.1)	0 (0.0)	1 (0.1)	
CCI score ≥ 5 (%)	460 (58.7)	73 (72.3)	387 (56.7)	0.003
Heart failure (%)	100 (12.8)	16 (15.8)	84 (12.3)	0.338
Myocardial infarction (%)	38 (4.9)	4 (4.0)	35 (5.1)	0.807
Asthma (%)	36 (4.6)	5 (5.0)	31 (4.5)	0.8
Kidney diseases (%)	151 (19.3)	17 (16.8)	134 (19.6)	0.589
Peptic ulcer (%)	36 (4.6)	2 (2.0)	34 (5.0)	0.303
Liver diseases (%)	26 (3.3)	8 (7.9)	17 (2.5)	0.016
COPD (%)	24 (3.1)	1 (1.0)	23 (3.4)	0.348
DM (%)	143 (18.3)	21 (20.8)	122 (17.9)	0.491
Brain hemorrhage (%)	90 (11.5)	8 (7.9)	82 (12.0)	0.314
Brain infarction (%)	171 (21.8)	24 (23.8)	147 (21.6)	0.607
Hemiplegia (%)	28 (3.6)	2 (2.0)	26 (3.8)	0.564
Dementia (%)	65 (8.3)	10 (9.9)	55 (8.1)	0.561
Connective tissue diseases (%)	37 (4.7)	5 (5.0)	32 (4.7)	0.805
Cancer (%)	134 (17.0)	13 (14.0)	111 (17.4)	0.226

Note: BMI, body mass index; CCI, Charlson Comorbidity Index; SD, standard deviation; COPD, chronic obstructive pulmonary disease; DM, diabetes mellitus; FIM, Functional Independence Measure.

**Table 4 ijerph-19-10125-t004:** Diagnosed conditions and their frequencies in admitted patients.

Medical Condition			Orthopedic Condition		
Diagnosis	Number	Percentage	Diagnosis	Number	Percentage
Brain stroke	139	44.3%	Femoral fracture	201	42.9%
Brain hemorrhage	75	23.9%	Compression fracture	122	26.0%
Pneumonia *	21	6.7%	Knee osteoarthritis	50	10.7%
Pyelonephritis	15	4.8%	Pelvic fracture	34	7.2%
Dehydration	12	3.8%	Spinal canal stenosis	15	3.2%
Other infections *	12	3.8%	Hip osteoarthritis	11	2.3%
Heart failure	11	3.5%	Patella fracture	11	2.3%
Cancer *	7	2.2%	Tibial fracture	9	1.9%
Intracranial hypertension	4	1.3%	Spinal cord injury	7	1.5%
Guillain–Barre Syndrome	4	1.3%	Radius fracture	4	0.9%
Aortic dissection	3	1.0%	Neck fracture	2	0.4%
Hernia	3	1.0%	Amputation	1	0.2%
Autoimmune diseases *	3	1.0%	Brachial plexus injury	1	0.2%
Bowel obstruction	2	0.6%	Clavicular fracture	1	0.2%
Epilepsy	2	0.6%			
Pulmonary embolism	1	0.3%			

* Pneumonia includes bacterial, viral, and aspiration pneumonia. Other infections include cholecystitis, cholangitis, septic arthritis, cellulitis, diverticulitis, appendicitis, psoas abscess, tuberculosis, and hepatic abscess. Cancer includes colon, stomach, pancreas, bone, and brain cancers. Autoimmune diseases include rheumatoid arthritis, polymyalgia rheumatica, and temporal arthritis.

**Table 5 ijerph-19-10125-t005:** Results of the multivariate logistic regression model regarding the relationship between dysphagia and home discharge.

Factor	Odds Ratio	95% CI	*p*-Value
Presence of dysphagia	0.38	0.20–0.73	0.0032
Age	1	0.97–1.03	0.85
Males	1.37	0.78–2.41	0.28
Albumin	1.09	0.69–1.70	0.72
Polypharmacy	0.53	0.32–0.90	0.018
Reasons for admission, internal medicine	0.44	0.26–0.77	0.0041
FIM score at admission			
Motor domain score	1.07	1.05–1.08	<0.001
Cognitive domain score	1.00	0.96–1.04	0.9
Dependent condition	1.26	0.74–2.17	0.4
CCI score ≥ 5 (%)	1.62	0.90–2.93	0.11

Note: CCI, Charlson Comorbidity Index; CI, confidence interval; FIM, Functional Independence Measure.

## Data Availability

The datasets used and/or analyzed during the current study may be obtained from the corresponding author upon reasonable request.

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
