# Peer review of "Relationship between Dysphagia and Home Discharge among Older Patients Receiving Hospital Rehabilitation in Rural Japan: A Retrospective Cohort Study"

_ijerph, 2022, doi:10.3390/ijerph191610125_

Round 1

Reviewer 1 Report

Thank you for submitting your paper for review and for conducting the study. Overall, this is a relevant topic where the objective was to clarify the relationship between the presence of dysphagia among older patients and discharge to their home from a rural rehabilitation unit. As the world's population is ageing rapidly, the lengthening of life expectancy puts at risk the adequate care of older chronically ill patients.  In this sense, a study that contributes to understanding ways to improve care for people with dysphagia at home is welcome.

The manuscript is well-written, and the English language is accurate both linguistically and regarding scientific purposes. I do not feel qualified to judge about the English language and style.

Here is a brief report of my assessment:

Title indicates the study’s design with a commonly used term.

Abstract presents an informative and balanced summary of what was done and what was found.

Introduction

Authors explain the scientific background and rationale for the investigation being reported, and state specific objectives.

Methods

Authors present key elements of study design in appropriate way; they give the eligibility criteria, and the sources and methods of selection of participants. Variables of study are clearly defined, and authors explain how quantitative variables were handled in the analyses.

Statistical methods are being described, and how missing data were addressed are explained.

A minor consideration in this section, the authors explain that patients who provided informed consent were included (lines 170-171), however, further down on lines 177-179, they explain that the study did not involve obtaining consent from participants because the data had already been collected. Please could you explain this aspect more clearly?

Results

Participant considerations are, in general, well explained and a flow diagram is presented.

Authors give characteristics of study participants, and they indicate number of participants with missing data for each variable of interest.

Main results are correctly presented.

A minor consideration in this section, I do not find that table 2 adds anything new to what is explained in the text (lines 277-281), it is advisable not to make the information redundant. Because of this, I suggest that you delete one of the two modes.

Discussion

In my opinion, a good analysis is presented here. The authors summarize the key findings with reference to the objectives of the study.

Limitations of the study are discussed with consideration of potential sources of bias or imprecision. The authors provide an overall interpretation of the results considering the objectives, limitations, multiplicity of analyses, results of similar studies and other relevant evidence.

Also, the external validity of the study results is discussed.

The conclusions are supported by the results.

Author Response

Responses to the reviewers’ comments

Thank you for reviewing our manuscript and providing suggestions for its improvement. We have provided point-by-point responses to the reviewers’ comments. Our revisions are indicated in red font here and in the manuscript. We hope that the revised manuscript meets the journal’s requirements and can now be considered for publication.

Thank you for submitting your paper for review and for conducting the study. Overall, this is a relevant topic where the objective was to clarify the relationship between the presence of dysphagia among older patients and discharge to their home from a rural rehabilitation unit. As the world's population is ageing rapidly, the lengthening of life expectancy puts at risk the adequate care of older chronically ill patients.  In this sense, a study that contributes to understanding ways to improve care for people with dysphagia at home is welcome.

The manuscript is well-written, and the English language is accurate both linguistically and regarding scientific purposes. I do not feel qualified to judge about the English language and style. 

Here is a brief report of my assessment:

Title indicates the study’s design with a commonly used term.

Abstract presents an informative and balanced summary of what was done and what was found.

Response:

Thank you for your valuable feedback.

Introduction

Authors explain the scientific background and rationale for the investigation being reported, and state specific objectives.

Response:

Thank you for your valuable feedback.

Methods

Authors present key elements of study design in appropriate way; they give the eligibility criteria, and the sources and methods of selection of participants. Variables of study are clearly defined, and authors explain how quantitative variables were handled in the analyses. 

Statistical methods are being described, and how missing data were addressed are explained.

Response:

Thank you for your valuable feedback.

A minor consideration in this section, the authors explain that patients who provided informed consent were included (lines 170-171), however, further down on lines 177-179, they explain that the study did not involve obtaining consent from participants because the data had already been collected. Please could you explain this aspect more clearly?

Response:

Thank you for your valuable feedback. Per your comment, we have deleted the description (Line 177 to 179).

Results

Participant considerations are, in general, well explained and a flow diagram is presented. 

Authors give characteristics of study participants, and they indicate number of participants with missing data for each variable of interest.

Main results are correctly presented. 

Response:

Thank you for your valuable feedback.

A minor consideration in this section, I do not find that table 2 adds anything new to what is explained in the text (lines 277-281), it is advisable not to make the information redundant. Because of this, I suggest that you delete one of the two modes.

Response:

Thank you for your valuable feedback. Per your comment, we have deleted the explanation of Table 2.

Discussion 

In my opinion, a good analysis is presented here. The authors summarize the key findings with reference to the objectives of the study. 

Response:

Thank you for your valuable feedback.

Limitations of the study are discussed with consideration of potential sources of bias or imprecision. The authors provide an overall interpretation of the results considering the objectives, limitations, multiplicity of analyses, results of similar studies and other relevant evidence. 

Also, the external validity of the study results is discussed. 

Response:

Thank you for your valuable feedback.

The conclusions are supported by the results.

Response:

Thank you for your valuable feedback.

Reviewer 2 Report

In this manuscript (ijerph-1832804), the authors investigated the relationship between dysphagia and discharge to home in older patients. The results that the presence of dysphagia, polypharmacy and admission for internal medicine diseases were negatively associated with home discharge are reasonable, but I have some concerns about this MS:

Totally, this manuscript is too long including many parts unrelated to this study’s results. It is like a review article. The authors should narrow the range of discussion about this study’s results. The part of direct discussion about this study’s results seems to be only 4.4 “Factors not related to home discharge among older patients.” In 4.3 of discussion, they mentioned drugs to treat depression, insomnia and delirium, but this study did not investigate the content of medication. They also cited the interprofessional collaboration in 4.5 and the consistency of food and the temporal flat lateral position in 4.6, but these have no direct relationship to the results.

The authors emphasize that the results were characteristic of rural Japan. But they did not compare the data in rural areas with the one in urban areas. It is not reasonable to refer to the characteristic of rural areas based only on the results.

Did the authors divide the participants into presence and absence of dysphagia based on the results of the water swallowing test? If it is so, it is not necessary to mention VE or VF in methods.

By citing references, the authors can simplify the information about the CCI and the FIM and omit the Table 2.

The beginning of the discussion, “Summary of the results”, is nothing but a repeat of the results. The authors can also omit this part.

In the fourth paragraph in 4.4 “Factors not related to home discharge among older patients” of discussion, the authors described as follows, “this study’s participants’ motor FIM with dysphagia was 71 on average, showing that they are frail.” Can I find the number “71” in Table 3? Do they have a clear rationale to take the score “71” as frail? Please add references to cite evidence, if it is so.

The conclusions are also too long. Please focus solely on what the authors can say based on the results.

Author Response

Responses to the reviewers’ comments

Thank you for reviewing our manuscript and providing suggestions for its improvement. We have provided point-by-point responses to the reviewers’ comments. Our revisions are indicated in red font here and in the manuscript. We hope that the revised manuscript meets the journal’s requirements and can now be considered for publication.

In this manuscript (ijerph-1832804), the authors investigated the relationship between dysphagia and discharge to home in older patients. The results that the presence of dysphagia, polypharmacy and admission for internal medicine diseases were negatively associated with home discharge are reasonable, but I have some concerns about this MS:

Totally, this manuscript is too long including many parts unrelated to this study’s results. It is like a review article. The authors should narrow the range of discussion about this study’s results. The part of direct discussion about this study’s results seems to be only 4.4 “Factors not related to home discharge among older patients.” In 4.3 of discussion, they mentioned drugs to treat depression, insomnia and delirium, but this study did not investigate the content of medication. They also cited the interprofessional collaboration in 4.5 and the consistency of food and the temporal flat lateral position in 4.6, but these have no direct relationship to the results.

The authors emphasize that the results were characteristic of rural Japan. But they did not compare the data in rural areas with the one in urban areas. It is not reasonable to refer to the characteristic of rural areas based only on the results.

Response:

Thank you for your valuable feedback. We agreed with the reviewer’s suggestions. Per your comment, we have comprehensively revised the discussion part by shortening the contents and connecting the contents with the research results and previous rural studies. (Lines 375 to 611)

“4. Discussion

For effective discharge to home, approaches to dysphagia should be considered in older patients, considering polypharmacy and ADL, as this study shows. Due to the negative effect of polypharmacy, the number and composition of medicines used by the patient should be noted upon admission, with the aim of reducing the number of medicines prescribed, especially in the internal medicine category. Moreover, based on the present and previous research, improving and sustaining ADL during admission may be essential for effective rehabilitation that leads to home discharge [30,41,42].

4.1 Dysphagia Impinges on Home Discharge of Older Patients in Rural Areas

Dysphagia leads to the deterioration of the patient’s medical condition. This research shows that dysphagia was related to difficulties in discharging patients to home. Within the hospital, patients with dysphagia needed special care around food consumption, with support from medical staff such as nurses, therapists, and care workers. Patients who cannot consume enough food by swallowing may need feeding using a transient tube to ensure proper nutrition [44]. Their rehabilitation quality can decrease in the rehabilitation period, and they become frailer than patients who do not have dysphagia [42,43]. This frailty can cause deterioration in health conditions and slow down their improvement [42,43].

Dysphagia also increases the burden on home medical care staff and caregivers, which directly impacts their ability to return home. The support needed for dysphagia can result in a substantial burden on home medical care staff [44]; for example, preparing and giving food to patients with dysphagia requires specific knowledge and skills [45]. Although home medical care staff are trained to provide this type of support, the burden associated with caring for individuals with dysphagia is still substantial and can directly affect individuals’ home discharge [46]. In rural areas, the number of home care professionals is limited. Thus, few home care workers are available to sufficiently support patients with dysphagia in their homes. Furthermore, home caregivers are not accustomed to providing such support [46]. Previous studies have shown that the burden on caregivers providing support for dysphagia can be significant in a home care setting [47]. Caregivers have to learn methods of providing support for dysphagia [46,47]. In this study, the discharge place was decided through discussion between medical professionals, patients, and home care givers. Workload and mental stress on the home caregiver can influence the home discharge rate among older patients in this study’s results.

Furthermore, the lack of a home workforce—such as families, relatives, and community workers—in rural areas can prevent home discharge among rural older patients. The continuity of home care for rural older individuals requires extensive family as well as social and welfare support. However, the rate of nuclear families has increased, and more elderly people live by themselves in rural areas, markedly so in Japan [48, 49]. Older individuals have to perform various activities independently [49]. If older individuals in rural areas have diseases that require hospital admission, they have to recover sufficiently because there may not be family members nearby to provide support at home [50,51].

4.2 Medicines Affecting Dysphagia in Internal Medicine

Various medicines can cause dysphagia, especially drugs such as those used to treat depression and insomnia. This research shows the possibility that several medicines can be associated with dysphagia. A previous study carried out in rural contexts showed the negative effects of medicines for mental illness on discharge to home [22]. The association can be explained by symptoms and diseases in the elderly. Many older patients have depression because of the symptoms of multiple diseases [52]. Some of them need medicines for symptoms of depression, such as selective serotonin reuptake inhibitors (SSRIs) and tricyclic antidepressants (TCAs) [53,54]. These medicines can affect the patients’ swallowing abilities because of dry mouth and drowsiness [55]. As saliva and being alert are essential for swallowing, patients using these medicines may have difficulty swallowing. Furthermore, sleep medicine for insomnia can affect their swallowing because of drowsiness [56]. Medicines for depression and insomnia can cause and exacerbate dysphagia in older admitted patients.

Medicines for delirium can also pose a risk for dysphagia. Delirium refers to a serious disturbance in psychosocial abilities that causes confused thoughts and reduced understanding of the environment [57]. Older patients have a high risk of delirium after ad-mission because their aged brains can be vulnerable to environmental changes [57]. In this study, the participants were mostly over 80 years old and had a high risk of delirium during hospital admission. For controlling symptoms such as agitation, insomnia, and hallucination, treatment with behavioral medicine is required [58]. When the treatment is not effective, SSRIs, TCAs, and benzodiazepine are used, which can cause dysphagia as a side effect [58]. Thus, admission causing delirium can be a risk factor for dysphagia in older patients.

Older patients admitted under internal medicine tend to have multimorbidity and require comprehensive care from various professionals and multiple medicines to control their symptoms. This approach to multimorbidity can cause polypharmacy, relating to dysphagia [59,60]. Furthermore, these patients may become delirious, as multimorbidity and polypharmacy are risk factors for delirium, causing dysphagia [61]. This study also shows that admission for the reason of internal medicine could impinge on discharge to home. To moderate the risk of polypharmacy and delirium, the involvement of various medical professionals is vital.

4.3 Factors not Related to Home Discharge Among Older Patients

Age might not be related to home discharge among older admitted patients. This study shows that aging is not associated with home discharge. This study focused on older generations, and their differences in physical and cognitive abilities may not differ. Previous studies have shown that aging could change people’s physical and cognitive functions although among older people, the speed of the change is not straightforward and depends on their lifestyles and environments. In addition, this study’s population is approximately 80 years old, so the change in their functions could vary, and the factor of age may not show a direct significant effect on discharge to their home from the hospital.

The difference in sex could not be related to home discharge. As this study shows, sex difference did not contribute significantly to home discharge. Sex difference could affect various clinical courses on diseases and behaviors. Sex differs in lay and professional care. In lay care usage such as self-management, consulting with families and friends, and over-the-counter drugs, there is no difference in the frequency of care between men and women [62]. In contrast, the usage of emergency and primary care may be different between men and women. Men are associated with infrequent visits to emergency departments, utilizing lay care more often to manage their symptoms [63]. Meanwhile, women tend to be associated with frequent usage of emergency departments [64]. These trends can be associated with the stereotype in some countries that men should be strong and masculine [63,64]. While men may handle their symptoms by themselves during emergencies, both men and women use primary care equally. Moreover, in the primary care setting, there are studies with different results involving usage rates between men and women who have critical symptoms [64,65]). In particular, older people tend to depend on primary care in rural settings. They use primary care frequently even when presenting with mild symptoms [66-68]. In rural contexts, primary care physicians could find older people’s critical symptoms and refer them to general hospitals for further investigation and hospitalization. Therefore, sex differences may not negatively affect their clinical courses of critical diseases in communities and hospitals.

This study shows that serum albumin may not be related to home discharge. Generally speaking, physicians and other medical professionals consider various clinical factors related to the discharge of older admitted patients in community hospitals effectively. Serum albumin levels may be associated with emergency admission to hospitals among older patients. Serum albumin levels may also be associated with infection, dehydration, and nutritional deficiency [4,69]. Low serum albumin levels indicate poor nutritional conditions with the inability to secrete albumin from the liver or severe infections with an increase in immunoglobulin-suppressing albumin production in the liver, which can cause high mortality [4,69]. In contrast, regarding older generations over 80 years old, there is a lack of evidence regarding the relationship between nutritional levels and home discharge. Their home discharge could be strongly affected by social conditions and their physical abilities. Therefore, this study may not show the relationship between home discharge and serum albumin levels.

This study clarified the contribution of motor FIM to discharge to home care, and cognitive FIM was related to their discharge to home. To decide upon the suitability of discharge to home, not only the change of motor FIM after rehabilitation but also basic physical ADL after the onset of diseases could be related [22]. At home, older people must sustain life functions regarding ADL, which are enhanced through rehabilitation. Cognitive functions may also affect older patient lives in their home to manage their usual lives. This research shows that the average cognitive FIM component was 32, higher than the suggested standard of living at home [34]. The study shows no relationship between discharging to homes and cognitive FIM statistically. Moreover, this study’s participants’ motor FIM with dysphagia was 61 on average, showing that they are frail [19,25]. More than 50% of them could go back to their home from the hospital. This research reveals a high possibility of older frail patients returning to their homes. Among older patients with cognitive dysfunction, their motor FIM's improvement enables them to discharge to their home. This study's participants are collected only from rehabilitation wards. Future studies can clarify the cause-and-effect relationship between improvement of the FIM and dysphagia, and discharge to home among older patients in hospitals.

In this study, the dependent conditions of the participants were not related to home discharge among the participants. Generally speaking, patients with dependent conditions have various kinds of diseases and may be exposed to high risks of acute conditions in their homes, which may impinge on their lives at home [70]. Medical staff consider that patients with dependent conditions should be cared for intensively. As their activities may not be intensive, they may not have many opportunities to develop symptoms, such as chest pain, dyspnea, and joint pain [71]. Additionally, as they may not be exposed to a range of people except medical staff, they may not frequently have communicable diseases [70]. Instead, less dependent patients may show symptoms according to their activities. As they may have certain problems expressing their symptoms, the various expression of symptoms may make their care givers anxious at home [48]. In contrast, the ability to express symptoms could facilitate early detection of critical symptoms, which can facilitate effective home care of less dependent patients.

This study shows that CCI may not have a statistical relationship with home discharge. Past medical history can be associated with an individual’s medical usage, triggered by symptoms from their diseases. People with specific chronic diseases and high CCI tend to use professional care because they may be accustomed to the usage of medical care for their frequent acute symptoms [72]. They can approach medical care quickly, allowing their medical conditions to be treated before becoming critical. In contrast, patients with low CCI may be able to control their symptoms with self-management [67] and may try to manage their symptoms with usual self-care. When they have symptoms with alarming signs, they may access medical care quickly, possibly leading to effective medical care. Therefore, high and low CCI could contribute to quick and effective medical care, and this study may not show the relationship between CCI and home discharge.

4.4 Comprehensive Methods to Address Dysphagia Among Older Patients

The approaches to the issues of dysphagia in older patients may need multiple interventions, including controlling multimorbidity and reducing medicines. This study’s results show that the effective control of various diseases and related prescriptions of multiple medicines may aggravate the problem of dysphagia, causing difficulty for discharge to home. The approaches to multimorbidity require considerable knowledge and skills in medical care. Comprehensive approaches including various professionals should be established for effective interventions.

Controlling multimorbidity in admitted older patients with acute diseases requires patient assessment by various professionals. In this study, the support for patients with dysphagia was performed only by therapists, nurses, and care workers. In patient care, physicians must diagnose patient diseases effectively and treat them with appropriate medicine and interventions involving patient and family participation [73]. Nurses and pharmacists can assess patient conditions in the context of patient discharge, considering the need for home care and reducing medicines [74,75]. In particular, pharmacists can assess older patients’ medicines in terms of polypharmacy and suggest which medicines can be reduced in admission situations [76]. Moreover, social workers and care managers can consider concretely how to care for older patients in their homes based on the long-term care insurance system [77]. As found in this study, older patients who were admitted may suffer from various diseases and need adjustment of medicine and care in their home. The collaboration among all these professionals can improve patient care, including dysphagia, and enable them to live in their homes.

Reducing polypharmacy to prevent the deterioration of swallowing in older patients should involve collaboration among multiple professionals. Based on the assessment of polypharmacy by pharmacists, physicians can choose which medicines can be reduced [78]. As this study shows, polypharmacy may affect the discharge destination from hospitals for elderly patients. These approaches can mitigate the negative effect of multiple medicines, leading to better conditions for the rehabilitation of swallowing by reducing the possibility of medicine-related complications [78].

4.5 Effective Methods of Eating for Patients with Dysphagia

Various swallowing methods can be applied for the continuation of oral food intake. First, the consistency of food can be changed according to the ability to swallow. Based on this study, there are many patients with brain stroke and hemorrhage who might have poor swallowing abilities because of neurological abnormalities. Orthopedic patients may also become frail owing to the limitation of their movement because of fractures [2,3], and their frailty can cause dysphagia [2,3]. Changing the consistency of food to a semi-solid or liquid form can enable patients to consume their food smoothly, lowering the risk of aspiration [79].

Second, the temporal flat lateral position (TFLP) has been introduced for the continuation of food swallowing. In this research, we assessed dysphagia based on VE and VF. These methods are usually performed in supine and reclining positions. Recently, TFLP has been used in VE and VF to find more effective ways for dysphagia patients to eat. The TFLP widens the space for food collection in the pharynx. This widening can sustain the food in the pharynx, leading to the prevention of aspiration while eating [80]. The usual space for collecting food in the pharynx is approximately 3 cm3. Conventional eating positions can allow the mass of food to enter the pharynx, with the effect of gravity causing aspiration. In contrast, the TFLP can widen the space to 15–20 mL without being affected by gravity, thereby reducing the risk of aspiration [80]. By using the TFLP, older patients with severe dysphagia and those who cannot consume food in conventional ways may be able to eat normally. The TFLP has been shown to improve mortality and hospital discharge rates [80]. This method could be effective even in rural hospitals and rehabilitation units [80]. As there are new methods of eating for patients with dysphagia, future studies should investigate comprehensive approaches in these patients, including education on new swallowing methods for medical professionals, patients, and their families.

4.6 Limitations

This study was performed at a single rehabilitation center in a rural Japanese hospital, and this might have affected the external validity. Future studies should investigate the effect of dysphagia on home discharge in older patients admitted to different types of hospitals in other countries. Nevertheless, this study can be used as a foundation for the investigation of rural rehabilitation regarding dysphagia.

Second, the excluded patients could have influenced the results. Excluded patients who died during admission may have had more severe conditions. BMI indicates a patient’s nutritional condition and is usually measured upon admission. Patients who lacked these data could have been more dependent because they were possibly unable to stand upright for their weight and height to be measured to calculate their BMI. Moreover, patients without albumin concentration data could have been in good nutritional condition because this measurement may not have been taken owing to their nutritional status. Therefore, the elimination of these participants may have affected the nutritional statistics in this study. Furthermore, care level and the duration of rehabilitation may have affected rehabilitation intensity. Patients without these data may have been more independent and, hence, may not have required assessment in terms of care level and may have been discharged to home without challenges. Thus, the reliability of the results of this study may have been affected. However, the number of excluded participants was relatively low; hence, the reliability ought to be retained.”

Did the authors divide the participants into presence and absence of dysphagia based on the results of the water swallowing test? If it is so, it is not necessary to mention VE or VF in methods.

Response:

Thank you for your valuable feedback. We agree with the reviewer’s comment. The explanation of the diagnosis of dysphagia is vague. Per your comment, we have revised the explanation of the diagnosis of dysphagia in the section on material and methods as follows. (Lines 205 to 218).

“The presence of dysphagia was defined based on the diagnosis by the otorhinolaryngologist using a water swallowing test (WST) score of three points or less. All patients were assessed by nurses for their abilities to swallow using the WST for the screening of dysphagia. The method used to diagnose dysphagia and perform swallowing rehabilitation was established based on the results of the videoendoscopic test (VE) and video-fluoroscopic examination (VF). If the WST score was three or less, the patients were assessed by VE performed by an otorhinolaryngologist to establish the dysphagia diagnosis and the rehabilitation procedure. If there were delays in swallowing or dysfunction of swallowing with no confirmed aspiration, speech therapists rehabilitated patients. If there was confirmed aspiration, VF was performed to determine what food consistency the patients could eat safely. Nurses and care workers helped patients with dysphagia to eat their food. Two groups of patients were distinguished by the presence or absence of dysphagia based on VE or VF.”

By citing references, the authors can simplify the information about the CCI and the FIM and omit Table 2.

Response:

Thank you for your valuable feedback. We agree with the reviewer’s comment. Per your comment, we have deleted the description and added the reference to FIM. We have left Table 2 for the explanation of FIM as follows. (Lines 271 to 284)

“The FIM is used for the assessment of basic ADL in patients in various situations. The FIM score is related to home discharge. The FIM consists of 18 items and is categorized into two components: motor and cognition. The scores for the two components are calculated based on the seven grades in each subscale. The motor subscales consist of eating, grooming, bathing, dressing the upper body, dressing the lower body, toileting, bladder management, bowel management, transfers to bed/chair/wheelchair, transfers to the toilet, transfers to the bath/shower, walking/using a wheelchair, and using the stairs. The cognition subscales consist of comprehension, expression, social interaction, problem solving, and memory. Each item is scored on seven original scales using a score of 1 to 7 (Table 2). A higher score indicates a more independent status. The range of motor FIM score is 13 to 91. The cognitive FIM score ranges from 5 to 35 [41].”

The beginning of the discussion, “Summary of the results”, is nothing but a repeat of the results. The authors can also omit this part.

Response:

Thank you for your valuable feedback. We agree with the reviewer’s comment. Per your comment, we have omitted the section of the summary of the results.

In the fourth paragraph in 4.4 “Factors not related to home discharge among older patients” of discussion, the authors described as follows, “this study’s participants’ motor FIM with dysphagia was 71 on average, showing that they are frail.” Can I find the number “71” in Table 3? Do they have a clear rationale to take the score “71” as frail? Please add references to cite evidence if it is so.

Response:

Thank you for your valuable feedback. I agree with the reviewer’s comment. Per the comment, we have added the reference and description regarding the frailty and FIM as follows. (Line 503-511)

“Moreover, this study’s participants’ motor FIM with dysphagia was 61 on average, showing that they are frail [19,25]. More than 50% of them could go back to their home from the hospital. This research reveals a high possibility of older frail patients returning to their homes. Among older patients with cognitive dysfunction, their motor FIM's improvement enables them to discharge to their home. This study's participants are collected only from rehabilitation wards. Future studies can clarify the cause-and-effect relationship between improvement of the FIM and dysphagia, and discharge to home among older patients in hospitals.”

The conclusions are also too long. Please focus solely on what the authors can say based on the results.

Response:

Thank you for your valuable feedback. I agree with the reviewer’s comment. Per your comment, we have shortened the conclusion section as follows. (612-619)

“The presence of dysphagia can negatively affect the discharge to home among patients admitted to a rehabilitation unit in a rural Japanese hospital. This relationship is a critical issue for aging societies in terms of sustaining comprehensive care for older people. The presence of dysphagia may be negatively associated with home discharge as are polypharmacy and admission for diseases treated under the umbrella of internal medicine. For effective discharge to home, approaches to alleviate dysphagia should be considered for rural older patients.”

Round 2

Reviewer 2 Report

I checked modifications of 2.6.2 and 2.6.3.4 of Methods, 4.3 and the front of Discussion, and the end of Conclusions. But I could not find where the authors changed Discussion because the whole text of Discussion was indicated in red font. I asked to narrow the range of Discussion about this study’s results and dramatically cut the parts of Discussion that are not directly connected with the results. I could find no trace of narrowing the discussion. Is every part of Discussion from 4.1 to 4.5 essential for the article? Please review again and drastically narrow down Discussion.

Author Response

Responses to the reviewer’s comments

Thank you for reviewing our manuscript and providing suggestions for its improvement. We have responded to the reviewer’s comments. Our revisions are indicated in red font in the manuscript. We hope the revised manuscript meets the journal’s requirements and can now be considered for publication.

I checked modifications of 2.6.2 and 2.6.3.4 of Methods, 4.3 and the front of Discussion, and the end of Conclusions. But I could not find where the authors changed Discussion because the whole text of Discussion was indicated in red font. I asked to narrow the range of Discussion about this study’s results and dramatically cut the parts of Discussion that are not directly connected with the results. I could find no trace of narrowing the discussion. Is every part of Discussion from 4.1 to 4.5 essential for the article? Please review again and drastically narrow down Discussion.

Response:

Thank you for the effective feedback on our manuscript. We agree with the reviewer's comments. We reviewed the entire discussion content and omitted redundant parts from the contents. Red color parts were the specific revised parts. In addition, I have added the reasons why we left each section.

4.1 Dysphagia Impinges on Home Discharge of Older Patients in Rural Areas

This section described the relationship between dysphagia and home discharge by referring to evidence related to home discharge in various contexts. In addition, we described the factors related to caregivers' burden in support of patient eating-related in this study's population and rural contexts. Following the reviewer's comment, we omitted some parts of the section.

4.2 Medicines Affecting Dysphagia in Internal Medicine

This section described the relationship between dysphagia and medicine by referring to evidence related to polypharmacy related to our research results. We described the relationship between polypharmacy and dysphagia by using articles showing evidence of the relationships. Following the reviewer's comment, we omitted some parts of the section.

4.3 ADL Affecting discharge to home

This section described the relationship between ADL and home discharge. This section was included in the following section, but we noticed this result was a positive relationship. So, we moved this section below section 4.2.

4.4 Factors not Related to Home Discharge Among Older Patients

This section was previous 4.3, which was reviewed by the reviewer, describing factors unrelated to home discharge. Following the reviewer's comment, we omitted some parts of the section.

4.5 Comprehensive Methods to Address Dysphagia Among Older Patients

This section describes the methods to overcome the issues of dysphagia among older patients by adding evidence and this research results. Although this section may not be strongly related to the research results, this section is important as a suggestion for future studies. So, we left this section for this reason. However, the content had a lot of redundant phrases. So, following the reviewer's comment, we omitted some parts of the section.